# Genome sequence of segmented filamentous bacteria present in the human intestine

Hans Jonsson[1]✉, Luisa W. Hugerth [2], John Sundh[3], Eva Lundin[4] & Anders F. Andersson [5]✉

Segmented filamentous bacteria (SFB) are unique immune modulatory bacteria colonizing the small intestine of a variety of animals in a host-specific manner. SFB exhibit filamentous growth and attach to the host's intestinal epithelium, offering a physical route of interaction. SFB affect functions of the host immune system, among them IgA production and T-cell maturation. Until now, no human-specific SFB genome has been reported. Here, we report the metagenomic reconstruction of an SFB genome from a human ileostomy sample. Phylogenomic analysis clusters the genome with SFB genomes from mouse, rat and turkey, but the genome is genetically distinct, displaying 65–71% average amino acid identity to the others. By screening human faecal metagenomic datasets, we identified individuals carrying sequences identical to the new SFB genome. We thus conclude that a unique SFB variant exists in humans and foresee a renewed interest in the elucidation of SFB functionality in this environment.

[1] Department of Molecular Sciences, Swedish University of Agricultural Sciences, 75007 Uppsala, Sweden. [2] Center for Translational Microbiome Research, Department of Molecular, Tumour and Cell Biology, Karolinska Institutet, 17177 Stockholm, Sweden. [3] Department of Biochemistry and Biophysics, National Bioinformatics Infrastructure Sweden, Science for Life Laboratory, Stockholm University, 17121 Stockholm, Sweden. [4] Medical Biosciences, Pathology, Umeå University, 90185 Umeå, Sweden. [5] Department of Gene Technology, Science for Life Laboratory, KTH Royal Institute of Technology, 17121 Stockholm, Sweden. ✉email: hans.jonsson@slu.se; anders.andersson@scilifelab.se

The interdependence of the intestinal microbiota and its host manifests itself in various ways, of which some on the host side are highly spectacular. Thus, effects spanning from improving nutrient uptake[1] or metabolizing drugs[2] to influencing risk of cancer[3] and altering cognitive function[4] have been reported to be dependent on microbiota composition and functionality. Although this area of research has attracted great attention during the last decades, the codes for communication between microbes and humans have only begun to be deciphered. Investigations of intestinal host–microbe interactions have been revolutionized by the development and application of powerful DNA sequencing and bioinformatics tools. Nevertheless, although huge amounts of data are gained in this way, the translation of these data into meaningful context lingers, and only a limited number of commensal bacteria has so far been identified as having defined effects on their host[5–8]. Segmented filamentous bacteria (SFB) represent such key players and hold a so far unique capacity to elicit full maturation of the mouse gut immune barrier. The work with SFB during the last decades beautifully describes the cross-fertilization between different areas of research, particularly microbiology and immunology[9].

SFB were discovered already in the mid-1960s in laboratory animals[10,11] where they could be identified by microscopy due to their filamentous growth and the unique attachment of the filament to the intestinal wall. Several intriguing features connected to the lifestyle of these organisms later opened for a deeper interest in their possible role as important symbionts[12]. Thus, they colonized primarily in the terminal part of the small intestine where many immune cells are located, they appeared at greater number around weaning which is an important period for maturation of the immune system, and, not least, they exhibited an intimate contact with the host through a specific anchoring to the intestinal cell wall. Together, these observations led to speculations and later also the first reports that SFB affected immune functions of the host[13,14]. After these early observations, SFB have been subject to a large number of studies (reviewed in[15,16]) which have firmly established their role as immunomodulatory bacteria. Thus, they are attributed with a wide range of effects, including stimulation of chemokine and antimicrobial components production, induction of gut lymphoid tissue and a strong increase in fecal IgA[17]. However, their potent triggering of T helper 17 ($T_H17$) cell differentiation is perhaps their most eye-catching attribute[18] in terms of immunomodulation. Interestingly, very recent experiments applying immunodeficient mice, demonstrated the ability of SFB to also confer protection against rotavirus infection independently of immune cells[19].

Laboratory work with SFB has been hampered by the fact that they are not yet cultivable as isolates, although methods of isolating[20] and co-culturing SFB from rodents with epithelial cells[21] have been described. Instead, SFB mono-colonized laboratory animals have offered the main route to characterization of this group of organisms. Complete genomes are available from SFB isolated from mice[22–25] and rats[24] and an unpublished draft genome sequence from turkey is publicly available (GenBank accession number GCA_001655775.1). Genomic analysis has revealed that SFB are Gram-positive spore-forming bacteria with a distinct phylogenetic position within the Clostridiales. They have small genomes of around 1.5 Mb which is reflected by a limited biosynthetic capability, rendering them a functional position between free-living bacteria and obligate intracellular symbionts[23,25].

A hallmark of SFB biology seems to be host specificity, as supported both by experimental and genetic data. Thus, colonization experiments have shown that cross-colonization with mouse SFB in rats or rat SFB in mice is not possible[26]. This argues for different species or lineages in SFB adapted to different hosts.

After the discovery of SFB in rodents and with the accumulating evidence for their ability to affect crucial steps in immune development, it was natural to search for them also in humans. The first study indicating their presence in humans visualized a tentative SFB organism adherent to ileal biopsied tissue by light microscopy[13]. This was later repeated in samples from patients with ulcerative colitis[27]. More recently, 16S rRNA gene sequences of SFB were reported in human samples using SFB-specific PCR primers; Yin et al.[28] found SFB sequences in 55 fecal samples while Jonsson[29] detected an SFB sequence in an ileostomy sample. The 16S sequences reported by Yin et al.[28] were phylogenetically interleaved with SFB sequences from mice and additional data from Chen et al.[30,31] also indicate that SFB with close relatedness to mouse SFB is present in humans. However, human gut shotgun metagenomic sample sets have been scanned by attempting to map reads to the SFB genomes of mice and rats, but without success[23]. In contrast, the 16S sequence described by Jonsson[29] from the ileostomy sample was distinct from SFB sequences from mouse and other animals. Although accumulating data suggest that SFB are present in humans, up until now no genomic sequence has been presented for human-derived SFB. Such a genome would provide a foundation for investigating SFB–host interactions in the human gut ecosystem.

We now report the draft genome sequence of a tentatively human-adapted representative of the SFB group. With metagenomic approaches, we have reconstructed the SFB genome from the same ileostomy sample that earlier produced the unique 16S rRNA gene sequence. Phylogenetic analysis clusters this genome to the SFB genomes described earlier, although it is genetically distinct from those. In addition, we could show the presence of sequences derived from the new genome in unrelated individuals through screening of published metagenome data. Our data strengthen the likelihood that the paradigm with host-specific colonization is valid also for SFB-human symbiosis. Considering the possibility of analogous immune-modulatory activities of SFB in humans and rodents, this finding could be of paramount importance.

## Results and discussion

**Genome reconstruction.** To verify the presence of an SFB 16S rRNA gene sequence in the human ileostomy sample where it was earlier detected with SFB-specific primers, we subjected the same sample to amplicon sequencing using broad-taxonomic range PCR primers. This confirmed the existence of an SFB sequence: after sequence noise removal, a single amplified sequence variant (ASV) was classified as 'Candidatus Arthromitus' and this was identical over its full length to the previously published 16S sequence from the same sample. The relative abundance of this ASV was however low, as it represented 0.16–0.37% of the microbial community's ASV sequences, depending on the DNA extraction method used.

In order to assemble the genome of the candidate SFB organism, we conducted deep shotgun metagenomic sequencing using Illumina NovaSeq, which generated a total of 953,167,834 read pairs for four different DNA preparations from the same sample. The 317,687 contigs of the resulting assembly were binned into genomes using information on sequence composition and coverage. To improve the binning procedure, the coverage of the contigs was estimated not only using the four different DNA libraries from the sample that were prepared using three different DNA extraction methods, but also using publicly available human gut metagenomes. We tried two different binning software, CONCOCT and MetaBAT2, and applied two different contig length cutoffs for each binning software. The two binners generated approximately the same number of bins with comparable quality estimates (Supplementary Fig. 1), but only

MetaBAT2 generated a bin at each length cutoff that was classified as SFB (genus *Savagella*[32] according to the Genome Taxonomy Database (GTDB)). These two bins differed by a few contigs, and we used a conservative approach of defining the SFB metagenome-assembled genome (MAG) as all contigs shared by both bins (127 contigs, 1,221,164 bp), as well as those uniquely found in one but taxonomically classified as SFB ('Candidatus Arthromitus' according to NCBI; 25 contigs, 89,165 bp). As is often the case for MAGs, contig(s) encoding a 16S rRNA gene were missing. rRNA gene prediction however identified a 4.2 kb contig encoding a 16S gene with a region identical to our SFB amplicon sequence, and this contig could be linked to contigs of the MAG using read-pair information. The contig (k141_89555) encodes a full-length 16S gene as well as a 23S gene. The 16S gene is 96% similar across its full length to those encoded in the SFB mouse and rat genomes. Notably, it has mismatches to commonly used primers for PCR identification of SFB in humans (Supplementary Fig. 2). Adding this contig resulted in a 1,314,549 bp (153 contig) MAG, that we denote SFB-human-IMAG (IMAG; ileostomy MAG). SFB-human-IMAG has a GC content of 26.98% and single-copy gene-estimated completeness and contamination of 85.6% and 0%, respectively (Supplementary Table 1).

**Phylogenomic analysis**. The reconstructed genome was subjected to phylogenomic analysis using a set of universally conserved protein sequences. This verified the placement of SFB-human-IMAG among the SFB (with 100% support). Intriguingly, the human-assembled SFB genome was most closely related to SFB isolated from turkey (GCA_001655775.1) and the two formed a sister clade to the SFB genomes from mouse and rat (Fig. 1). This pattern was supported by average amino acid identity (AAI) analysis, with SFB-human-IMAG displaying 71% AAI to SFB turkey, while displaying 65% AAI to the SFB from rodents (Table 1). It was however not supported by a phylogenetic tree based solely on the full-length 16S genes of the genomes (Supplementary Fig. 3). The conflicting phylogenies between the SFB and their hosts could indicate that the SFB have switched hosts during the course of evolution. It could also reflect that the human and turkey SFB belong to a different lineage than the mouse and rat SFB, and that the two lineages diverged before mammals diverged from birds. The two SFB lineages may exist in all hosts, or one could have gone extinct in some of them.

To put the sequence similarity of the SFB genomes into perspective, prokaryotes displaying 65% AAI usually belong to the same family, but in most cases (ca. 60%) to different genera, while at 71% AAI they typically (ca. 70%) belong to the same genus, based on the NCBI taxonomy[33]. Compared to the host genomes, 71% AAI is on par with the AAI between orthologous proteins of the chicken and human genomes (75.3%)[34], while 65% is considerably lower than the 85% AAI between orthologs of human and mouse[35]. However, vertebrate host genomes have been reported to evolve more slowly than their bacterial symbionts[36]. Compared to other gut bacteria inhabiting multiple host species, such as *Escherichia coli* and *Lactobacillus reuteri*[37,38], SFB-human-IMAG displays a much lower similarity to its relatives in the other hosts. However, the other bacteria display clear signs of having inhabited or switched hosts after the host species diverged, and, to the best of our knowledge, there are no commensal gut bacteria with sequenced genomes to compare with that are known to have stably inhabited their hosts since primates diverged from avians or even from rodents.

**Gene content and physiology**. SFB-human-IMAG encodes 1276 proteins. Clustering these proteins together with proteins from

SFB genomes from mice, rat, and turkey formed 2222 protein clusters, of which 904 have representatives in all genomes, 428 are specific to rodents, 305 are specific to turkey, and 219 are specific to human SFB (Fig. 2, Supplementary Fig. 4 and Supplementary Data 1). Fifty-two clusters were shared exclusively by human and turkey SFB. Annotating the protein clusters with COGs[39] showed that the percentage of genes assigned to different COG functional categories did not differ markedly between SFB-human-IMAG and the other SFB genomes (Supplementary Fig. 5), indicating that the organisms have overall similar functional capabilities. The 271 protein clusters found in SFB-human-IMAG but lacking in rodents were significantly enriched in the COG functional category V (Defense mechanisms) and category X (mobilome: prophages, transposons) and in proteins lacking COG annotation (Fisher's exact test, false discovery rate adjusted *P* value < 0.05; Supplementary Fig. 5). SFB-human-IMAG lacks 227 protein clusters that were found in all the other SFB. Many of these are likely missing due to the genome being incomplete.

SFB have previously been described as having fermentative metabolism. We have identified all but a few of the enzymes for glucose utilization also in the draft genome of SFB-human-IMAG. No enzymes involved in the tricarboxylic acid cycle were identified, and, accordingly, there are no proteins that can be assumed to take part in an electron transport chain, confirming a fermentative lifestyle. As previously described SFB, SFB-human-IMAG appears to have a restricted capability to synthesize amino acids, vitamins/cofactors, and nucleotides. One interesting observation, however, is that SFB-human-IMAG contains six genes for biotin synthesis (BIOA, B, D, F, W, and X), suggesting it has the capability to synthesize biotin. The corresponding genes show sequence homology to genes from SFB-turkey but most of them are lacking in the SFB from rodents. This could reflect differences in the physiology or diet of the hosts, or differences in the microbial community in which rodent and human SFB exist, since some gut microbes can synthesize this cofactor while others are auxotrophs. Nine glycoside hydrolases (GH) representing six different GH families (Supplementary Table 3), one tentatively extracellular N-acetylglucosaminidase, and several cell surface-bound and extracellular proteases were identified in the draft genome. Together, these enzymes are likely used for harvesting components from the intestinal milieu.

The SFB-human-IMAG genome also encodes a large number of transport functions. This is in agreement with a restricted metabolic capability and similar to other SFB. Since the genome is not complete, some transport functions are likely missing due to incomplete genome assembly. A notable exception is the lack of the ABC transporter for phosphonate, where the specific genes are missing in the middle of an SFB-human-IMAG contig that otherwise displays conserved synteny with SFB-mouse-Japan. However, since phosphorous is indispensable, bacteria have evolved several systems for acquisition of this macronutrient, and SFB, including SFB-human-IMAG, carry genes for a phosphate specific transport system (sfb.merged_01113—sfb.merged_01115).

When comparing with the annotation of the complete genome of SFB-mouse-Japan, we conclude that SFB-human-IMAG is likely to carry a complete set of genes for sporulation and germination. Likewise, a complete set of genes for flagellar motility and chemotaxis are present, and it is thus reasonable to assume that the bacterium has the ability for motility and chemotaxis.

**Host–microbe interactions**. The intestinal microbiota influences the host locally or systemically through a number of mechanisms. These could be of an indirect character such as the production of

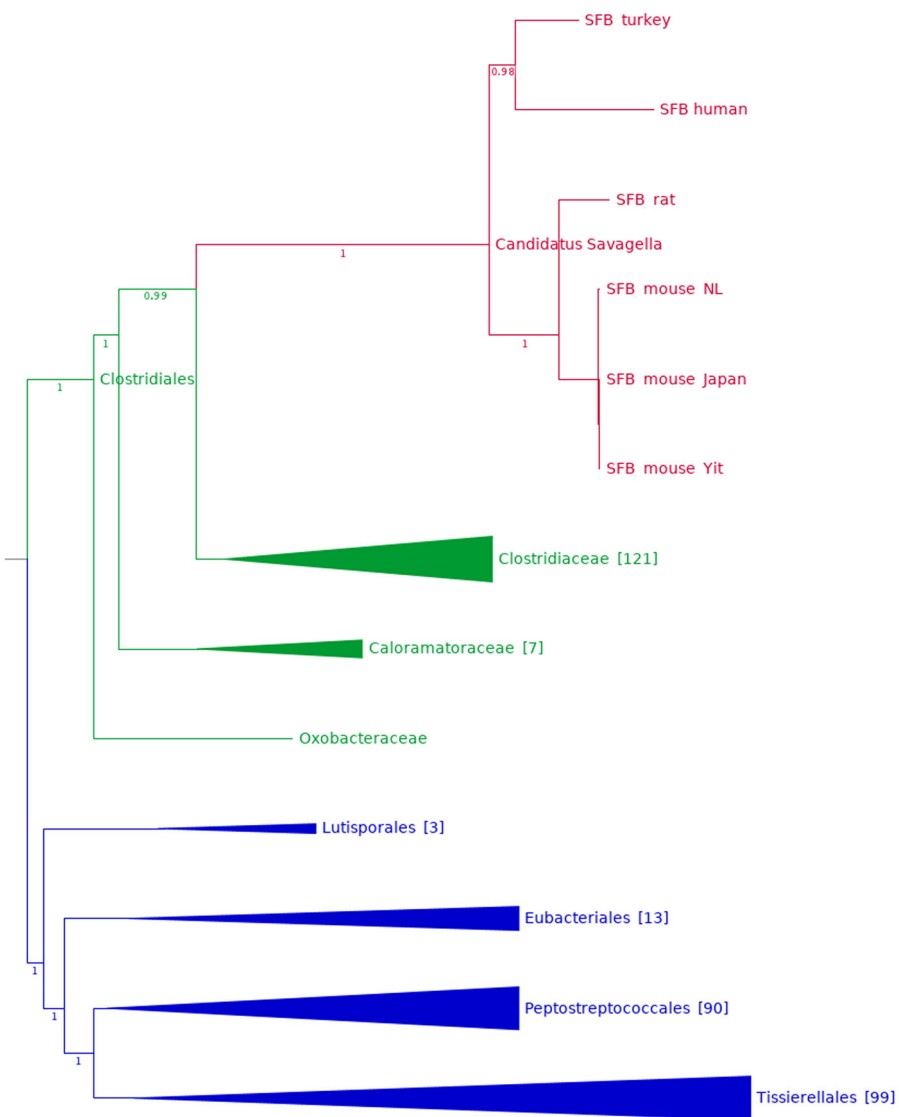

**Fig. 1 Phylogenomic tree of genome-sequenced SFB and related Clostridia.** The genus '*Candidatus* Savagella' (family Savagellaceae) is highlighted in red; other families of order Clostridiales are depicted in green. Internal branches are marked with support values (range 0–1). Orders that form a monophyletic sister group to Clostridiales are shown in blue.

**Table 1 Average amino acid identity between sequenced SFB genomes.**

|                     | SFB-rat-Yit | SFB-mouse-Japan | SFB-mouse-Yit | SFB-mouse-NL | SFB-turkey-UMNCA01 |
|---------------------|-------------|-----------------|---------------|--------------|--------------------|
| SFB-mouse-Japan     | 83.52       |                 |               |              |                    |
| SFB-mouse-Yit       | 83.51       | 99.78           |               |              |                    |
| SFB-mouse-NL        | 83.52       | 98.89           | 98.96         |              |                    |
| SFB-turkey-UMNCA01  | 69.32       | 69.49           | 69.5          | 69.51        |                    |
| SFB-human-IMAG      | 65.31       | 65.15           | 65.12         | 65.34        | 70.57              |

vitamins or other metabolites that interact with host functions. The intestinal microbiota as a whole is responsible for such effects, and thus, specific interactions, and individual microbial contributions may be concealed. More direct effects could be mediated by bacteria that physically interact with host structures. While the bulk of our knowledge about such interactions comes from studies of various pathogens, the increasing knowledge of SFB biology has the potential to change this. These bacteria interact with their host in a way not seen with other commensal/symbiotic bacteria and a number of proteins and functions of SFB

have been proposed as instrumental in host association and immune-modulatory effects. Among these are immunogenic flagellins, tentative fibronectin binding proteins which could effectuate binding to host cell matrix, phospholipase C, and ADP ribosyltransferase, both influencing actin polymerization which is a characteristic at sites of SFB attachment, and others[22–24,40]. Due to the prevailing limitations in culture and genetic manipulation, however, very limited experimental data pinpointing the role of individual SFB components in host interactions are available. One example though, is the ability of SFB flagellins to interact in vitro

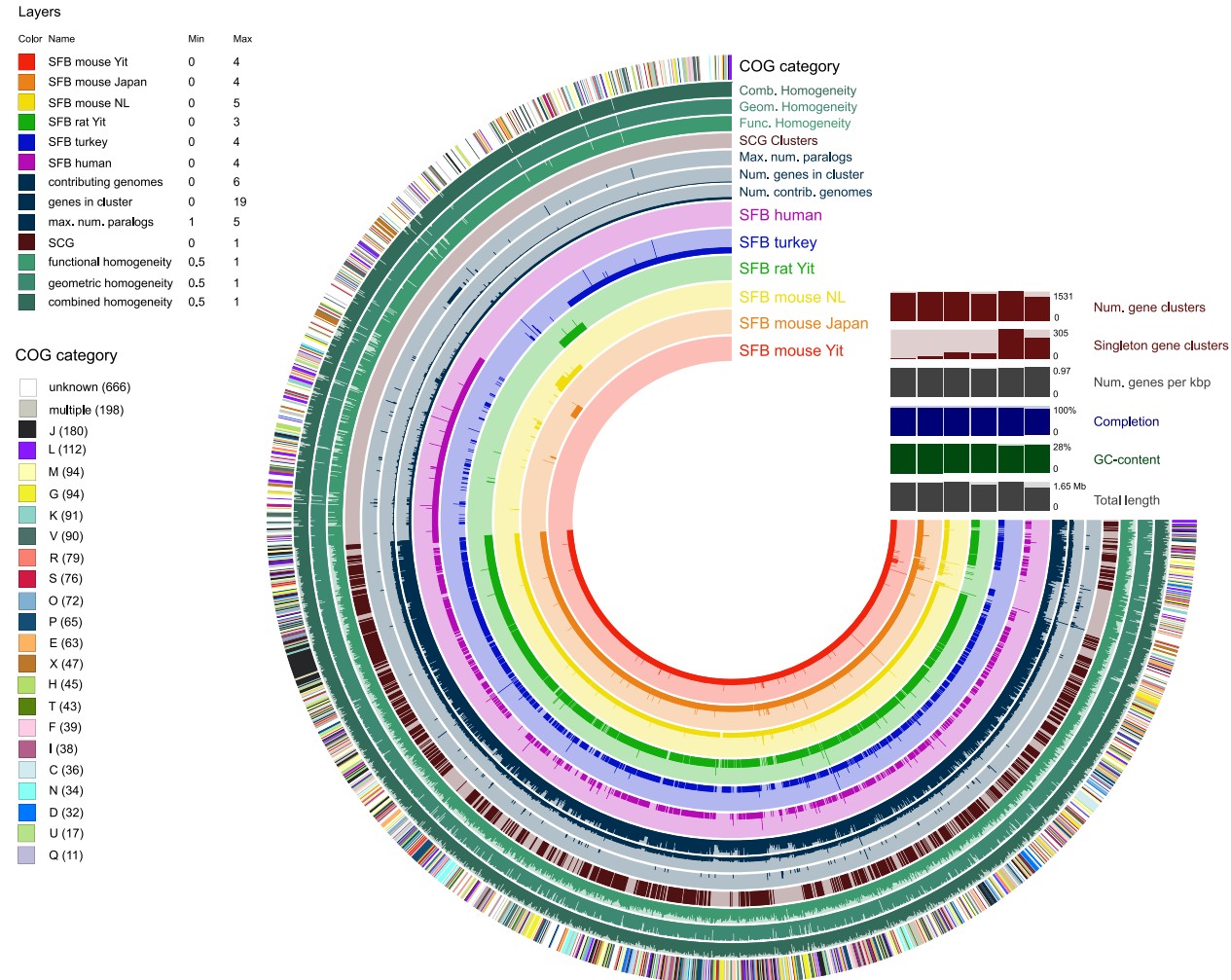

**Fig. 2 Pangenomic analysis of SFB genomes.** Pangenome graph generated with Anvi'o, where the gene clusters (radial bars) are ordered according to the organization of genes in the SFB-mouse-Yit genome. The circles show, from inner to outer, for each gene cluster, the number of gene copies in each genome (circles 1–6); the number of genomes with the gene cluster present (circle 7); the number of genes in the cluster (circle 8); the maximum number of gene copies among the genomes (circle 9); single-copy gene (SCG) clusters (present once in each genome)(circle 10); functional- (circle 11); geometric- (circle 12); and combined homogeneity of the gene cluster (circle 13); COG category (circle 14). The homogeneity reflects the conservation of the protein sequences within a gene cluster.

with TLR5 receptors and to activate the NF-κB signaling pathway and elicit the innate immune response[22]. We identified three flagellins in the genome of SFB-human-IMAG and at least two of these contain a conserved motif for TLR5 recognition and activation[41] (Supplementary Fig. 6).

SFB interacts physically with intestinal enterocytes through polar attachment of the SFB-filament and triggers an invagination of the enterocyte without breaching the cell membrane. This intimate contact suggests a strong potential for interaction with the host, and indeed, data were recently published that show how SFB in mice can transfer cell wall proteins into the enterocyte[42]. This protein (p3340) was earlier shown to be a major target in the antigen-specific CD4 T$_H$17 cell response induced by SFB[43]. The corresponding protein is also encoded in the SFB-human-IMAG genome (sfb.merged_00774). It is interesting to note that while the N-terminal (signal sequence) and the C-terminal parts of these proteins display high amino acid identity, the main part shows only a low degree of identity (Supplementary Fig. 7). In the work by Yang et al.[43] two peptides from p3340 were reported to strongly stimulate T$_H$17 cells. These peptides are conserved only to a limited degree in SFB-human-IMAG, leaving open the possibility that the variability in sequence reflects host adaptation

and thus the evolvement of human-specific T$_H$17 triggering epitopes.

The components of SFB responsible for attachment to the enterocytes have not been identified. Secreted and cell surface located bacterial proteins generally play major roles in signal transduction, ion transport and host cell adhesion. While enzymes and transporters often contain signature motifs, many proteins involved in adhesion are undefined as to their functional sites, and therefore depicted as hypothetical. A number of secreted and cell surface proteins were predicted in our genome based on N-terminal signal peptides and 60 of these are hypothetical proteins. The size of the hypothetical and tentatively extracellular proteins in SFB-human-IMAG ranges from 57 to 2040 amino acids, and the identity to homologous proteins from SFB from other animal hosts are in the range of 34–72%, with a mean of 52% identity. This is substantially lower than the overall identity of the SFB-Human-IMAG proteome with other SFB (Table 1), and thus indicates more rapid evolution in proteins communicating with the exterior environment. It is plausible that some of these proteins play a role in attachment and host communication and thereby mediate the host specificity that is a characteristic of SFB.

SFB do not harbor a gene for sortase, the enzyme that normally anchors many cell surface proteins in Gram-positive bacteria, and a corresponding mechanism has not been described in the SFB group. It is likely though that an alternative route for anchoring of cell surface proteins exists in SFB. Interestingly, a conserved amino acid motif located C-terminally was earlier identified in a number of putative cell surface proteins in SFB[40]. We have localized this motif in several extracellular proteins, including the $T_H17$ stimulating protein p3340 from mouse SFB and the orthologous protein 00774 from SFB-Human-IMAG mentioned above. Supporting evidence for anchoring comes from the study of Ladinsky et al.[42], where immuno-EM shows the location of p3340 to the SFB cell wall. We therefore postulate that SFB-human-IMAG has at least 14 cell surface proteins that may be anchored to the bacterial surface through the involvement of this aa-motif. Furthermore, 12 proteins encoded by the SFB-Human-IMAG genome are predicted to be anchored via a lipoprotein motif, and 6 proteins could possibly be anchored via an N-terminal transmembrane helix (TMHMM 2). This leaves a substantial number of predicted extracellular proteins seemingly anchorless. While some of these likely are true secretory proteins, it is notable that a number of them have a very high isoelectric point, giving them a basic charge which in turn could allow them to re-associate with the bacterial surface[44].

**Presence in other metagenomes**. To verify that SFB-human-IMAG resides in the human intestine, we searched for it among published metagenomes from the human gut. The genome was first BLAST-searched against an integrated catalog of reference genes in the human gut microbiome (IGC 9.9)[45], which consists of 9.9 million genes assembled from 1267 human fecal samples. Only 14 of the IGC genes gave matches to the genome when requiring ≥95% identity and ≥70% of the IGC gene's bases aligned. However, this gene catalog is mainly derived from samples from adults, while SFB in most animals peak in young individuals during weaning[46]. Therefore, we instead scanned a large recent metagenomic study consisting of a time series of fecal samples from children 0–3 years born in Russia, Estonia, and Finland[47]. The reads from the metagenome samples were first mapped against SFB-human-IMAG using standard settings. This rendered substantial mapping for many samples. However, manual inspection of the alignments revealed that the mapped reads were typically only partially aligned, and to regions displaying unusually high sequence conservation, such as structural RNA genes. Redoing the mapping with stringent settings (see 'Methods') and only counting reads mapped to protein-coding genes (CDS) gave substantially reduced mapping; however, 61 out of the 153 contigs were mapped by at least one read pair, and 7 out of the 817 samples had at least one read-pair mapping. Two of these samples, one Estonian infant at day 390 (SRS1719092) and one Finnish infant at day 320 (SRS1719390), had particularly many reads mapping and mapped with 1–3 read pairs each to 24 and 44 contigs, respectively. Although the SFB-mapping reads only corresponded to 3 and 11 out of a million mapped reads, respectively, in these samples, the reads appeared to be randomly distributed over the genome, indicating that the genome is present in these samples, rather than that some genome regions are wrongly binned or horizontally transferred. In comparison, zero reads from the 817 samples mapped to any of the rodent SFB genomes using the same settings, and the draft SFB genome from turkey was mapped only at two CDS, both corresponding to genes being 100% identical at the nucleotide level to genes of several Firmicutes genomes. Since the infant metagenomes analyzed were derived from another lab, it can be excluded that the mappings to SFB-human-IMAG are due to contamination of DNA from our Ileostomy sample or sequencing library. We also checked for the presence of SFB in the metagenomes from intestinal luminal fluids from three Chinese children that had earlier been screened positive for SFB with PCR[30]. With the exception for reads mapping to one of the above SFB-turkey CDS, no mapping to any of the SFB genomes were obtained for these samples. In summary, our analyses show that SFB-human-IMAG is present in human infant fecal material, although in very low relative abundance.

SFB hold a so far unique position in our collective knowledge on how individual components of the intestinal microbiota can affect host functions. The intimate interaction with the intestinal cells represents a remarkable evolutionary mechanism and recent data have shown that this is indeed a route for SFB–host interaction. Although SFB has been described from many host species, conclusive data regarding a human-specific SFB has been lacking. The data presented in this study strongly suggest that such a lineage actually exists. The assembled genome clusters with the previously described SFB genomes while being clearly distinct from these. The insight that SFB could be a natural component of the human microbiota calls for deepened attempts to elucidate their impact on human physiology in general and immune development in particular.

## Methods

**Sample collection and storage**. Samples were initially collected and processed as described by[48]. Briefly, ten adult subjects previously proctocolectomised for ulcerative colitis volunteered to participate in the experiment (two female subjects, eight males, age range 24–65 years, BMI 20.7–35.6 kg/m2). The subjects were living a normal life based on physical examination and blood tests before the experiment. The study was approved by the Ethical Committee of the Umeå University Hospital (approval number 89–102) in compliance with the Helsinki declaration. The participants actively participated in the original project by registration of the food intake and the deliveries of their ileostomy bags[48], and participants were informed that the samples were frozen for future measurements in various forms. Ileostomy bags were immediately frozen on dry ice and stored at −30 °C. Ileostomy effluents from each 24 h period were freeze-dried to constant weight, mixed, homogenized, and stored at −70 °C until analysis. One of the subjects was earlier[29] identified as positive for the presence of an SFB-related 16S rRNA sequence on the basis of PCR analysis and sequencing. Sample from this individual was used in this work.

**DNA extraction**. The DNA used for the 16S amplicon sequencing was extracted using QIAamp DNA Stool Mini Kit (QIAGEN, Venlo, The Netherlands) with an added bead beating treatment as the first step. Bead beating was performed with 0.1 mm zirconium/silica beads (Biospec Products, Bartlesville, OK, USA), 2 × 45 s with setting 5 using the MP FastPrep-24 (MP Biomedicals, Irvine, CA, USA). Of the five samples used for amplicon sequencing, the first two were extracted from the original material while the latter three corresponded to three size fractions, selected by gravity precipitation. Since the SFB content of these size fractions was not significantly larger than for the full sample, all later DNA extractions were performed on unfractionated material. For the shotgun sequencing, two replicates were extracted with QIAamp DNA Stool Mini Kit, one with QIAamp Fast DNA Stool Mini Kit, and one with QIAamp DNA Microbiome Kit, according to instructions from the manufacturer (QIAGEN, Venlo, The Netherlands).

**16S rRNA gene amplification and sequencing**. DNA extracts were amplified using universal 16S primers 341f and 805r[49] enhanced with Illumina adapters as described by[50] (341f: 5′-ACACTCTTTCCCTACACGACGCTCTTCCGATCT-N5-CCTACGGGNGGCWGCAG-3′; 805r: 5′-AGACGTGTGCTCTTCCGATCTGGA CTACHVGGGGTWTCTAAT-3′, where N5 represents five random bases used to improve sequencing quality) using 25 μl of Kapa Hifi mastermix (Kapa Biosystems, Woburn, MA, USA), 2.5 μl of each primer (10 μM), 2.5 μl of template DNA (1 ng/μl), and 17.5 μl of water. These mixtures were submitted to thermocycling in a Mastercycler Pro S (Eppendorf, Hamburg, Germany) under the following conditions: 95 °C for 5 min, 98 °C for 1 min, 20 cycles of 98 °C for 20 s, 51 °C for 20 s, and 72 °C for 12 s, followed by a final elongation step of 72 °C for 1 min. The products of these reactions were cleaned as described by Lundin et al.[51], concentrating the product to 23 μl. These were then barcoded in a PCR reaction containing 25 μl Kapa Mastermix polymerase and 1 μl of each barcoding primer (5′-AATGATACGGCGACCACCGAGATCTACAC-X8-ACACTCTTTCCCTACA CGACG-3 and 5′-CAAGCAGAAGACGGCATACGAGAT-X8-GTGACTGGAGT TCAGACGTGTGCTCTTCCGATCT-3′, where X8 is a barcoding sequence) and with the following cycling conditions: 95 °C for 5 min, 98 °C for 1 min, 10 cycles of 98 °C for 10 s, 62 °C for 30 s, and 72 °C for 15 s, followed by a final elongation step

of 1 min. The products were cleaned again as described by Lundin et al. and concentrated to 15 µl. DNA concentration was measured with Qubit dsDNA HS (Thermo Fisher Scientific, Waltham, MA, USA) and the length and purity of the amplified product was verified with BioAnalyzer 2100 DNA1000 (Agilent Technologies, Santa Clara, CA, USA).

The products were sequenced on Illumina MiSeq with $2 \times 300$ bp together with amplicon samples from a different project. Cutadapt v.1.18[52] was used to remove primer sequences, 3′-bases with a Phred score <15, and sequences not containing the expected primers. The resulting sequences were submitted to Unoise3[53]. Taxonomic annotation was performed with SINA based on SILVA 132[54].

**Metagenomic library preparation and sequencing**. Libraries were prepared with the ThruPLEX DNA-seq kit (Rubicon genomics, Ann Arbor, MI, USA), aiming at an average fragment length of 350 bp. Sequencing was performed in a NovaSeq 6000 in S1 mode, yielding 358–410 million reads/sample.

**Preprocessing of shotgun reads**. For the ileostomy samples, adapters were trimmed from the sequences using cutadapt[52] (v. 1.18) with default settings using the adapter sequences AGATCGGAAGAGCACACGTCTGAACTCCAGTCAC (ADAPTER_FWD) and AGATCGGAAGAGCGTCGTGTAGGGAAAGAGTGTAGATCTCGGTG GTCGCCGTATCATT (ADAPTER_REV). Removal of phiX sequences was performed by aligning reads against the phiX genome (GCF_000819615.1) using bowtie2[55] (v. 2.3.4.3) with parameters '--very-sensitive' and only keeping pairs that did not align concordantly. Duplicates were removed using fastuniq[56] (v. 1.1) with default settings. This was followed by a second cutadapt trimming step using parameters '-e 0.3 --minimum-length 31'. Reads were then classified taxonomically using kraken2[57] (v. 2.0.7_beta). Reads classified as human were removed prior to assembly.

Three external datasets of human gut samples were used for binning and for checking the presence of the obtained SFB MAG: 21 samples from BioProject PRJNA288044 (unpublished), 785 samples from BioProject PRJNA290380[47], and 11 samples from BioProject PRJNA299342[30]. The 21 PRJNA288044 samples and the 11 PRJNA299342 samples were preprocessed by adapter and quality trimming using Trimmomatic[58] (v. 0.38) with parameters 'PE 2:30:15 LEADING:3 TRAILING:3 SLIDINGWINDOW:4:15 MINLEN:31' followed by removal of phiX sequences as above. The 785 PRJNA290380 samples were preprocessed in the same manner but with the NexteraPE adapters and with duplicate removal following the phiX filtering step.

**Assembly and binning**. Preprocessed ileostomy shotgun reads were assembled using megahit[59] (v. 1.1.3) with settings '--min-contig-len 300 --prune-level 3 --k-list 21,29,39,59,79,99,119,141'. The resulting assembly consisted of 317,687 contigs, totaling 437,952,431 bp. The contig length distribution had min 300 bp, max 509,384 bp, avg 1379 bp, and N50 2118 bp. Preprocessed reads from all samples (ileostomy and external samples) were then aligned against the assembled contigs using bowtie2[55] (v. 2.3.4.3) with '--very-sensitive' settings, followed by duplicate removal using MarkDuplicates (picard v. 2.18.21[60]) with default settings. This output was used to calculate contig abundance profiles in all samples using the jgi_summarize_bam_contig_depths script from metabat2[61] (v. 2.12.1). Binning of assembled contigs was then performed in two runs using metabat2 with parameters '--seed 123 -m <min_contig_length>' where 'min_contig_length' was set to 1500 and 2500 for the two runs. For binning using CONCOCT[62] (v. 1.0.0), contig abundance profiles were computed using the concoct_coverage_table.py script followed by binning in two separate runs, both using default settings but with minimum contig length ('-l') set to 1000 and 2500, respectively. Bin quality was assessed using checkm[63] (v. 1.0.13) using lineage-specific marker genes. Ribosomal RNA genes were identified on assembled contigs using barrnap (https://github.com/tseemann/barrnap) (v. 0.9) with parameters '--reject 0.1'.

**Taxonomic annotation of contigs**. Assembled contigs were classified taxonomically using package tango (https://github.com/johnne/tango, v. 0.5.6) and the UniRef100 protein database (release 2019_02). The package queried contigs in a blastx search using diamond[64] (v. 0.9.22) with parameters '--top 5 --evalue 0.001'. From the results, contigs were assigned a lowest common ancestor from hits with bitscores within 5% of the best hit. Assignments were first attempted at species level using only hits at ≥85% identity. If no hits were available at that cutoff, an attempt was made to assign taxonomy at the genus level using hits at ≥60% identity, followed by the phylum level at ≥45% identity. These rank-specific thresholds were chosen from[65].

**Functional annotation of genome**. The SFB-human-IMAG bin as well as five sequenced genomes of SFB (RefSeq accessions GCF_000284435.1 GCF_000709435.1 GCF_000283555.1 GCF_001655775.1 GCF_000270205.1) were annotated using prokka[66] (v. 1.13.3) with default settings. The prokka pipeline includes tRNA identification with aragorn (v. 1.2.38), prediction of ribosomal RNA with barrnap (https://github.com/tseemann/barrnap) (v. 0.9), gene calling with prodigal[67] (v. 2.6.3), homology searching with blastp[68] (v. 2.7.1+), and HMM-profile searches with hmmer[69] (v.3.2.1). Protein sequences predicted with prokka

were further annotated using eggnog-mapper[70] (v. 1.0.3) in 'diamond' run mode with the 4.5.1 version of the eggNOG database. Kegg orthologs, enzymes, pathways, and modules were inferred from the eggnog-mapper output using the Kegg Brite hierarchy information. Proteins were also annotated with PFAM protein families using pfam_scan.pl (v. 1.6) with default settings and v. 31 of the PFAM database. Carbohydrate-active enzyme annotations were inferred using hmmscan against the dbCAN (http://bcb.unl.edu/dbCAN/) database (v. 6), followed by parsing of the output with the hmmscan-parser.sh script downloaded from the dbCAN server (http://bcb.unl.edu/dbCAN/download/hmmscan-parser.sh) and filtering using settings recommended for bacteria in the dbCAN readme ($E$ value < $1e^{-18}$ and coverage > 0.35).

SignalP-5.0 was used to identify signal peptides in the predicted proteins of SFB-human-IMAG. Organism group was set to Gram-positive.

**Pangenomic analysis**. The genbank files generated by Prokka for SFB-human-IMAG as well as for the other five SFB genomes were loaded into Anvi'o (v. 6)[71]. COG annotation was run in Anvi'o using blastp. Anvi'o's pangenome analysis[72] was run with default settings, except that blastp was used instead of diamond for cross-comparing the protein sequences for the clustering.

**Phylogenetic and amino acid similarity analyses**. The phylogeny of the SFB genomes was inferred using GTDB-TK[73] (v. 0.2.2) with GTDB release86, in both 'classify_wf' and 'denovo_wf' modes. The former placed the query genomes into an existing reference tree using pplacer[74] while keeping the reference tree intact and was used to assign a GTDB taxonomy to the genomes. The latter instead created a new phylogenetic tree using both reference and query genomes and was used to investigate the phylogenetic relationship between the genomes. In the 'denovo_wf' method FastTree[75] (v. 2.1.10) was used with the WAG protein model and Gamma20-based likelihoods ('-wag -gamma').

For the 16S phylogenetic analysis, one full-length 16S rRNA gene from each of the previously published complete SFB genomes, as well as from the genomes of five different species of Clostridium, were downloaded from the RDP[76]. The positioning of the 16S rRNA gene in SFB-human-IMAG contig k141_89555 and in SFB-turkey contig GCF.001655775_NZ_LXFF01000001.1 was predicted with CheckM. The six 16S genes were aligned with Muscle[77] and columns with gaps removed with DegePrime[50]. A phylogenetic tree was constructed with FastTree using the GTR + CAT model (results were nearly identical using the Jukes–Cantor + CAT model).

Average AAI between genome pairs were calculated using the online AAI calculator (http://enve-omics.ce.gatech.edu/aai/index), using default parameter settings.

**Quantifying SFB in external metagenomes**. Matching of the ORFs in IGC v9.9 (db.cngb.org/microbiome) against SFB-human-IMAG was performed with blastn v2.7.1+[68] requiring at least 80% identity over at least 70% of the query sequence. To assess the presence of SFB-human-IMAG and of SFB from mouse, rat, and turkey in the feces of young children, we used the recent work of Vatanen et al.[47], one of the datasets that we used for the binning. Mapping of the preprocessed reads against the SFB genomes was run in 'strict' mode, where only alignments without mismatches were reported ('--score-min C,0,0' in bowtie2). Counts of read pairs mapping inside protein-coding regions (CDS) was obtained with featureCounts[78] (v. 1.6.4) with settings '-p -B -M' to only count read pairs with both ends mapped and allowing multimapping reads. The same procedure was used for mapping the shotgun reads from Chen et al.[30].

**Statistics and reproducibility**. Analysis of enrichment of COG functional categories in SFB-human-IMAG relative to SFB from rodents was conducted with Fisher's exact test and using false discovery rate adjustment of $P$ values to account for multiple testing.

**Reporting summary**. Further information on research design is available in the Nature Research Reporting Summary linked to this article.

## Data availability
The preprocessed amplicon and shotgun sequencing reads generated during this study, all the contig sequences, as well as the contig sequences of SFB-human-IMAG, are available at the European Nucleotide Archive (ENA) under the study accession number PRJEB34939, where ERZ1468256 points to the contig sequences of SFB-human-IMAG. Data files for amplicon sequence variants, genome and gene sequences and annotations (Prokka, PFAM, eggNOG, dbCAN), phylogenomic analysis, genome quality estimates and metagenome read mappings are available[79]. Protein cluster information, including COG annotations, is available in Supplementary Data 1. All other data (if any) are available upon reasonable request.

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

## Acknowledgements
DNA sequencing was conducted at the Swedish National Genomics Infrastructure (NGI) and at Clinical Genomics at Science for Life Laboratory (SciLifeLab) in Stockholm. Computations were performed on resources provided by the Swedish National Infrastructure for Computing (SNIC) through the Uppsala Multidisciplinary Center for Advanced Computational Science (UPPMAX). We thank Göran Hallmans, Umeå University, for access to the ileostomi samples and Lars Engstrand, CTMR/KI, for his contribution to shogun sequencing discussions. Open access funding provided by Royal Institute of Technology.

## Author contributions
H.J. conceived the study. H.J. and A.F.A. designed the study. E.L. coordinated and conducted sampling. H.J. conducted DNA extractions. J.S., L.W.H., and A.F.A. conducted bioinformatics analyses. H.J., L.W.H., J.S., and A.F.A. interpreted data and wrote the paper. All authors approved the final version of the paper.

## Competing interests
The authors declare no competing interests.
