## [Peer Review File · Communications Biology]

Reviewers' comments:

Reviewer #1 (Remarks to the Author):

Segmented filamentous bacteria (SFB) is an important bacteria that have the function on regulating host immunity balance. It is hard to culture of SFB in vitro prevent the deep study of this bacteria. That is a good news if the human SFB genome sequences can be assembled and published. However, I think the convince of this study is not enough.

1. The quantity of SFB in each sample should be detected using real-time PCR and listed in supplementary.
2. 16S rRNA gene sequencing results should be listed in supplementary.
3. Why only the three external datasets were chose for analysis.
4. It would be better to present the distribution of the merged human SFB genome sequences on so many different human samples. It will be interesting to see how many contigs or sequences obtained by authors. If the merged sequenced combined from so many samples, I hesitated and worried the accuracy.
5. When compared to mouse and rat SFB, the similarity of human SFB at amino acid level is lower than 70%. That is may be need more explain.
6. Authors don't do any test to verify the accuracy. I think many details not provided by authors. Others will hard to duplicated this research.

Reviewer #2 (Remarks to the Author):

This manuscript makes a very important contribution, namely unequivocally demonstrating that Segmented filamentous bacteria (SFB) indeed exists in the human intestinal tract and moreover providing a reasonably complete genome that should pave the way for other studies to determine the extent to which plays an important role in human physiology. Some specific suggestions are offered.

While much of the text is geared for those with expertise in genome assembly, this paper will have broad interest by those without such expertise. Indeed, most new genomes don't merit a high-profile journal but SFB does, both for overcoming the challenge of assembling it from such low abundance and because of its outsized importance- at least in mice- suggesting possible great import in humans, a topic this work makes investigable. Anyway, I suggest some of the technical jargon be reigned in considering the audience.

I suggest the authors put the similarity of human SFB vs. other species SFB into perspective by discussing how this compares with other bacteria that colonize multiple species. E. coli would surely a start. Many clostridia strains can be found in various species. How similar are they?

Minor:

Please unify the citations format throughout the article, such as L106, L296, L352...

Obviously, data needs to be deposited into a mineable format once the work is accepted.

L66: " Segmented filamentous bacteria, SFB" is plural - change to " Segmented filamentous bacteria, SFB, represent".

L118, L120, L150: How did you determine uniqueness? Based only on percentage?

L118, what percentage does this strain of SFB accounts for of the total bacterial population in this ileostomy sample? At such percentage, how did you ensure the sequencing quality?

L138: Please provide relevant statistics for the assembly (N50 values).

L149: Please provide genome GC content of the genome.

L218 and elsewhere: How were the functions of the genes were determined? Annotation? Please briefly describe.

Reviewer #3 (Remarks to the Author):

The manuscript by Jonsson et al. describes the isolation of bacterial DNA from human ileostomy samples and construction of the genome of segmented filamentous bacteria (SFB). As a keystone species in the induction of several immune responses and the associated colonization resistance, this is a major advance in our understanding of human isolates of SFB. The bioinformatics were rigorous and conservative, as they should be, resulting in an almost complete assembly with minimal if any genomic contamination. Other strengths of the study include their phylogenomic and predicted functional comparisons to existing avian and rodent SFB genomes. Overall, the manuscript is fairly well-written. The main weakness of the manuscript is the insufficient citation of relevant sources. In more than one instance, the authors' assertions are based on incomplete references, all of which are described below alongside a couple of other minor concerns.

1. The authors state in line 89 that SFB are not yet cultivable, which is not accurate. This statement should be amended and the authors are advised to briefly review (or at least mention) the studies published by Ericsson et al. (2015) *BioTechniques*, 59(2):94-98 and Schnupf et al. (2015) *Nature*, 520:99-103 describing different methods of isolating and maintaining or even culturing SFB from rodent hosts.

2. Recommend capitalizing "Gram-positive" (line 93 and elsewhere) as Gram is a proper noun.

3. Regarding the description of evidence supporting a human-adapted SFB variant in the Introduction (lines 103 to 114), I would recommend the authors also include references by Chen et al. (2018) *Frontiers in Microbiology*, 9:1403 as well as the report from Caselli et al. (2013) *Am J Gastroenterol*, 108:860-861. Given these studies alongside the work from Yin et al (reference 25) and the authors themselves, the concluding remark for this paragraph (i.e., "it is still an open question whether a human-adapted variant of the organism actually exists") does not seem accurate. There is ample evidence now that human SFB exist, which is completely to be expected given the presence of cognate strains in almost every host species examined. I recommend re-wording this final sentence to emphasize what is really unknown – the genomic and molecular machinery by which SFB exert their immunomodulatory effects.

4. In the term "Candidatus *Arthromitus*", *Candidatus* should not be italicized as this simply denotes that this is a provisional (i.e., candidate) taxonomy. *Arthromitus*, on the other hand, should be italicized as this is the proposed Latin designation for a new genus.

Reviewers' comments:

Reviewer #1 (Remarks to the Author):

Segmented filamentous bacteria (SFB) is an important bacteria that have the function on regulating host immunity balance. It is hard to culture of SFB *in vitro* prevent the deep study of this bacteria. That is a good news if the human SFB genome sequences can be assembled and published. However, I think the convince of this study is not enough.

1. The quantity of SFB in each sample should be detected using real-time PCR and listed in supplementary.

Response: The quantity of SFB in the ileostomy sample has not been assessed by qPCR. However, based on the 16S amplicon sequencing data, we report the frequency of the SFB 16S ASV relative to all bacterial ASVs in the sample (0.16 - 0.37%; line 132), which we believe is more relevant than the number of DNA copies per volume of sample. The external metagenome samples where we detected the SFB-human-IMAG *in silico* we do not have access to.

2. 16S rRNA gene sequencing results should be listed in supplementary.

Response: These results are available in the accompanying Zenodo archive (<https://doi.org/10.5281/zenodo.3504269> - a link for access during the review process is provided in the manuscript) and the raw 16S sequencing data is available through ENA (study accession number PRJEB34939) as described in the Data availability section.

3. Why only the three external datasets were chose for analysis.

Response: We chose to search for SFB-human-IMAG in one of the largest metagenome studies on infants, the DIABIMMUNE project. This dataset consists of 785 samples and scanning this amount of data is already a large computational task. In follow up studies it would be interesting to analyse other datasets as well.

4. It would be better to present the distribution of the merged human SFB genome sequences on so many different human samples. It will be interesting to see how many contigs or sequences obtained by authors. If the merged sequenced combined from so many samples, I hesitated and worried the accuracy.

Response: The SFB-human-IMAG draft genome was obtained from one single ileostomy sample. After assembling the sequence reads from this sample into contigs, two binning runs were run with slightly different parameter settings. Each of these binning runs resulted in a bin classified as SFB. The SFB-human-IMAG draft genome was defined as all contigs shared by both bins (the majority of contigs), as well as those uniquely found in one bin but taxonomically classified as SFB. We refer to the assembly as "sfb.merged" because it is merged from the two binning runs.

5. When compared to mouse and rat SFB, the similarity of human SFB at amino acid level is lower than 70%. That is may be need more explain.

Response: This is a good suggestion. We have now added a new paragraph on lines 176-188 that addresses this. See also the end of the previous paragraph with relevance to this (lines 170-174).

6. Authors don't do any test to verify the accuracy. I think many details not provided by authors. Others will hard to duplicated this research.

Response: We are unsure what the reviewer means by “accuracy” in this context. The accuracy of the genome assembly and binning is tested using the well established software CheckM. This estimated SFB-human-IMAG to be 85.6% complete and having 0% contamination. According to the Minimum information about a metagenome-assembled genome (Bowers 2017, Nat Biotech), a Medium-quality draft MAG has a completion $\geq 50\%$ and contamination $< 10\%$, and a High-quality draft MAG a completion $> 90\%$ and contamination $< 5\%$. Thus, our MAG is considerably above Medium-quality and almost fulfills the criteria for a High-quality draft MAG.

Accuracy in terms of detecting the MAG in external metagenome samples, is according to us, very well described. We try to make it clear in the text that by using standard settings for mapping of metagenome reads, we do get quite a bit of false positives. Therefore, we take means to do the mapping with very stringent settings, but still we get reads mapping to SFB-human-IMAG from multiple samples. As negative controls, we also map to SFB genomes from rodents and turkey using the same settings, and get zero read mapping to the rodent genomes and only a few reads mapping to two CDS in the SFB turkey genome.

When it comes to providing details and the reproducibility of the study, we have described every bioinformatic step in detail, including parameter settings and versions of the bioinformatic softwares. All raw sequencing data (16S and shotgun), assembled contigs, and the SFB-human-IMAG bin are provided (ENA study accession PRJEB34939) and many analysis files are also available (with descriptions) at <https://zenodo.org/record/3504269#.XteccZ4zZD8>.

Reviewer #2 (Remarks to the Author):

This manuscript makes a very important contribution, namely unequivocally demonstrating that Segmented filamentous bacteria (SFB) indeed exists in the human intestinal tract and moreover providing a reasonably complete genome that should pave the way for other studies to determine the extent to which plays an important role in human physiology. Some specific suggestions are offered.

Response: Thank you for the positive comments!

While much of the text is geared for those with expertise in genome assembly, this paper will have broad interest by those without such expertise. Indeed, most new genomes don't merit a high-profile journal but SFB does, both for overcoming the challenge of assembling it from such low abundance and because of its outsized importance- at least in mice- suggesting possible great import in humans, a topic this work makes investigable. Anyway, I suggest some of the technical jargon be reigned in considering the audience.

Response: We agree that the Results section regarding the assembly and binning is rather detailed, but since reconstruction of the genome is a fundamental aspect of this manuscript, and since some scientists are still rather sceptical to metagenome-assembled genomes, we felt it is motivated to describe the procedure in a detailed and transparent manner.

I suggest the authors put the similarity of human SFB vs. other species SFB into perspective by discussing how this compares with other bacteria that colonize multiple species. E. coli would surely a start. Many clostridia strains can be found in various species. How similar are they?

Response: This is a good suggestion. We have now added a paragraph addressing this (lines 176-188).

Minor:

Please unify the citations format throughout the article, such as L106, L296, L352...

Response: This has now been fixed.

Obviously, data needs to be deposited into a mineable format once the work is accepted.

Response: The ENA accession number to the genome is now provided (SAMEA6614744 under study accession PRJEB34939). Data will be released by ENA any day (if not already).

L66: " Segmented filamentous bacteria, SFB" is plural - change to " Segmented filamentous bacteria, SFB, represent".

Response: This has now been fixed.

L118, L120, L150: How did you determine uniqueness? Based only on percentage?

Response: On L118, "the unique 16S rRNA gene sequence" refers to the preceding paragraph ("the 16S sequence described by Jonsson (29) from the ileostomy sample was distinct from SFB sequences from mouse and other animals").

On L120, based on that the genome is far from identical to the previously described SFB genomes (based on the tree and based on AAI). This has now been reformulated to “although it is genetically distinct from those” (line 119).

On L150, “uniquely found in one but taxonomically classified as SFB” refers to those contigs that were only included in one of the two bins. This is not based on percentage, but based on the contig IDs. (The same set of contigs was binned in two separate binning runs.)

L118, what percentage does this strain of SFB accounts for of the total bacterial population in this ileostomy sample? At such percentage, how did you ensure the sequencing quality?

Response: Based on the 16S amplicon sequencing data, it represents 0.16 - 0.37% of the community. The low relative abundance made it necessary to conduct very deep shotgun sequencing. One way that we lowered the risk of propagating errors introduced in the library preparation was to sequence from 4 independent sequencing libraries.

L138: Please provide relevant statistics for the assembly (N50 values).

Response: Complete assembly statistics have now been added on line 435-437: “The resulting assembly consisted of 317,687 contigs, totalling 437,952,431 bp. The contig length distribution had min 300 bp, max 509,384 bp, avg 1379 bp and N50 2118 bp”

L149: Please provide genome GC content of the genome.

Response: This has now been added to line 158.

L218 and elsewhere: How were the functions of the genes were determined? Annotation? Please briefly describe.

Response: We now added “Annotating the protein clusters with COGs” to line 206. Further information is provided in the Methods section.

Reviewer #3 (Remarks to the Author):

The manuscript by Jonsson et al. describes the isolation of bacterial DNA from human ileostomy samples and construction of the genome of segmented filamentous bacteria (SFB). As a keystone species in the induction of several immune responses and the associated colonization resistance, this is a major advance in our understanding of human isolates of SFB. The bioinformatics were rigorous and conservative, as they should be, resulting in an almost complete assembly with minimal if any genomic contamination. Other strengths of the study include their phylogenomic and predicted functional comparisons to existing avian and rodent SFB genomes. Overall, the manuscript is fairly well-written. The main weakness of the manuscript is the insufficient citation of relevant sources. In more than one instance, the authors' assertions are based on incomplete references, all of which are described below alongside a couple of other minor concerns.

Response: We are glad that the reviewer found our study an important contribution to the field and being rigorously conducted.

1. The authors state in line 89 that SFB are not yet cultivable, which is not accurate. This statement should be amended and the authors are advised to briefly review (or at least mention) the studies published by Ericsson et al. (2015) *BioTechniques*, 59(2):94-98 and Schnupf et al. (2015) *Nature*, 520:99-103 describing different methods of isolating and maintaining or even culturing SFB from rodent hosts.

Response: Thank you for bringing to our attention these two references. We have made a brief reference to the two articles suggested by changing the text to: “Laboratory work with SFB has been hampered by the fact that they are not yet cultivable as isolates, although methods of isolating (Ericsson et al. 2015) and co-culturing SFB from rodent hosts with epithelial cells (Schnupf et al. 2015) have been described.”

2. Recommend capitalizing “Gram-positive” (line 93 and elsewhere) as Gram is a proper noun.

Response: This has now been fixed.

3. Regarding the description of evidence supporting a human-adapted SFB variant in the Introduction (lines 103 to 114), I would recommend the authors also include references by Chen et al. (2018) *Frontiers in Microbiology*, 9:1403 as well as the report from Caselli et al. (2013) *Am J Gastroenterol*, 108:860-861. Given these studies alongside the work from Yin et al (reference 25) and the authors themselves, the concluding remark for this paragraph (i.e., “it is still an open question whether a human-adapted variant of the organism actually exists”) does not seem accurate. There is ample evidence now that human SFB exist, which is completely to be expected given the presence of cognate strains in almost every host species examined. I recommend re-wording this final sentence to emphasize what is really unknown – the genomic and molecular machinery by which SFB exert their immunomodulatory effects.

Response: We have now added the references by Chen et al. and Caselli et al. and reformulated the end of this paragraph (lines 101-113).

4. In the term “*Candidatus Arthromitus*”, *Candidatus* should not be italicized as this simply denotes that this is a provisional (i.e., candidate) taxonomy. *Arthromitus*, on the other hand, should be italicized as this is the proposed Latin designation for a new genus.

Response: According to <https://en.wikipedia.org/wiki/Candidatus>, “*Candidatus*” but not the subsequent name should be given in italics for provisional taxa. However, it is also recommended that the whole name is written within parentheses. This is now done.

REVIEWERS' COMMENTS:

Reviewer #2 (Remarks to the Author):

I am a bit surprised to learn that "there are no commensal gut bacteria with sequenced genomes to compare with that are known to have stably inhabited their hosts since primates diverged from avians or even from rodents." but I nonetheless thank the authors for looking for them.

Revised manuscript makes an important contribution.

Reviewer #3 (Remarks to the Author):

The authors have adequately responded to my suggestions.